# Debunking Divine Command Theory

Hans Van Eyghen 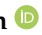

Tilburg School of Catholic Theology, Tilburg University, 5037 AB Tilburg, The Netherlands; hansvaneyghen@gmail.com

**Abstract:** The divine command theory holds that morality finds its origin in God or that God is somehow closely connected to morality. Many people across the world hold a related, though different belief that Religious belief is required for proper moral behavior. In this paper, I look at a number of evolutionary and cognitive explanations (supernatural punishment theory, big gods theory, moral dyad, and costly signaling) that purport to explain why people hold beliefs concerning a close connection between God and morality. I assess whether any of these theories provide a reason for epistemic concern.

**Keywords:** divine command theory; evolution of religion; cognitive science of religion; religion and morality; costly signaling; supernatural punishment theory; big gods theory; moral dyad

## 1. Introduction

The divine command theory is one of the best known meta-ethical theories. The theory comes in several forms which share a commitment that morality is closely connected to God. The divine command theory bears a resemblance to commonsensical beliefs regarding the origins or nature of moral norms.

Several theories from the cognitive science of religion have attempted to explain why and how people form beliefs regarding a connection between God and morality. Some seemingly support the idea that a close connection between God and morality or a belief that God is at the root of morality can be explained away as features of human psychology. In this paper, I assess whether these theories have any negative epistemic bearing on beliefs concerning the God-morality connection. Below, I argue that cognitive theories only have a negative epistemic bearing on beliefs in line with the divine command theory if theistic evolution is ruled out. I also argue that some cognitive theories do signal epistemic problems for the idea that belief (or faith) in God is required for proper moral behavior.

This paper only takes cognitive theories into account that are widely discussed and assumed as true by many contemporary scholars. It leaves out older, discarded theories or fringe theories. Regarding divine command theory, the paper relies on recent defenses thereof.

This paper is structured as follows: In Section 2, I discuss the main tenets of divine command theory and beliefs associated with the theory; in Section 3, I discuss four theories that attempt to explain beliefs regarding the God-morality connection; in Section 4, I assess the epistemic impact of these four theories. I end with a conclusion.

## 2. What Is the Divine Command Theory?

### 2.1. Varieties of Divine Command Theory

The divine Command Theory (DCT) is one of the main meta-ethical theories defended by meta-ethicists. In its classical form, it holds that morality is dependent on God's will or that something is good if and only if it was commanded by God (Austin 2006).[1] Proper moral behavior, therefore, consists of obeying God's commandments. God's commandments can be both positive (e.g., 'Honor your father and your mother' (Ex 20: 12a)) and

negative (e.g., 'You shall not murder' (Ex 20:13)). Lists of divine commandments are usually drawn from sacred scriptures.[2]

As Elisabeth Anscombe notes, DCT is an exemplar of a law-based conception of morality where proper moral behavior consists of following laws or rules (Anscombe 2020).[3] On DCT, God is the law-giver and humans ought to follow the laws. Unlike in the case of civil laws, following divine laws requires following the spirit of the law rather than the letter of the law.

Edward Wieringa distinguishes strong and weak versions of DCT. Strong versions claim that moral norms are defined in terms of divine commandments, e.g., x is commanded by God and, therefore, obligatory. Strong versions advance a dependency relation where moral norms are dependent on God's will. Weaker versions merely claim that the commands of God are coextensive with the demands of morality. On weaker versions, morality does not have its origins in God's commands but God's commands merely inform us about its content (Wierenga 1983).

### 2.2. Religious Belief as a Requirement for Morality

The DCT, like most meta-ethical views, is the domain of professional philosophers. Like most other meta-ethical views, it resembles more commonly held positions regarding morality. A lot of non-philosophers hold the belief that God is somehow at the root of morality. The idea of God issuing commandments that subsequently ground a moral system is clearly present in the Old and New Testaments and in other sacred scriptures. Adherents of other traditions also commonly believe in a transcendent origin of morality. For example, most Chinese traditions see morality as having its origin in a command or mandate from Heaven (Tian)[4] (Zhao 2009).

Empirical data also suggest a related common belief in a much stronger claim, which is sometimes disavowed by defenders of DCT.[5] Many people accept that belief in God is necessary for proper moral behavior or proper moral character. In one study (Tamir et al. 2020), the question 'Is belief in God necessary to be moral and have good values' was asked to a panel of 38,426 people. Overall, a small majority of 51% responded with 'yes' and 45% with 'no'. The ratio of affirmative answers was much lower in Western European countries (22%). A total of 33% of respondents from Eastern Europe answered affirmatively. In the USA, 44% answered yes and 26% did so in Canada. Affirmative answers were much more common outside of the Western world, with 96% in the Philippines and Indonesia, 79% in India, 53% in Korea, 75% in Turkey, and 95% in Kenya.[6]

Despite the objections by defenders of Divine Command Theory (DCT) regarding the necessity of religious belief for fostering proper moral behavior, it is noteworthy that certain similarities exist between the two claims. Notably, both systems emphasize a close linkage between morality and God. Whereas adherents of DCT merely point to God as the originator of moral norms, many accept the need for continued support for religious belief. Advocating for the need for religious belief to uphold proper moral conduct does not necessarily entail attributing the origin of morality directly to God. They, however, appear to occur together often.

### 2.3. Divine Command Theory and Belief

From the discussion above, we can distinguish a number of claims regarding the connection between God and moral norms.

1. Moral norms are good or bad solely because they were issued by God.
2. Moral norms stem from the nature of God.
3. God is somehow involved in morality.
4. Commands of God are coextensive with demands of morality.
5. Belief in God is required for proper moral behavior.

Claims (1) and (2) fit with stronger readings of the divine command theory. Claims (3) and (4) are in line with weaker readings. Claim (5) is often disavowed by defenders of DCT but commonly believed throughout the world (see above).

Claims 1–5 are, therefore, believed by a subset of the human population. These beliefs can be assessed epistemically in a number of ways. A straightforward way to assess 1–4 is by looking at the arguments in favor of DCT and criticism leveled against DCT. This falls beyond the scope of this paper.[7] A different way is looking at biases or similar belief-forming processes that often give rise to beliefs in line with 1–5. A close investigation may reveal that beliefs in line with 1–5 are formed in epistemically bad ways.

What does it mean for a belief to be formed in an epistemically bad way? On most accounts of reliabilism (cf. Goldman and Beddor 2021), a belief-forming process (BFP) is epistemically bad if it yields a high ratio of false beliefs. A clear example is wishful thinking. Such a standard is, however, not helpful for assessing BFPs that do not have clear truth to falsity ratios. The approach I will be taking below regards a BFP as epistemically bad if it misprocesses information. A clear example of a BFP that misprocesses information is looking while wearing bad glasses. Subjects who engage in this practice see a distorted view of reality where solid objects appear fluid, and persons appear doubled. They do not see reality as it is, but the BFP misleads them into perceiving things in a different way. If a belief originates from a BFP that distorts reality, epistemic worries for the belief arise. The worries are often sufficiently strong to doubt that the belief is justified or rational.[8]

Below, I look at evidence that may suggest that beliefs in line with 1–5 are often formed by BFPs that misrepresent or distort reality. A positive conclusion does not rule out that beliefs in line with 1–5 can be formed by other BFPs that are reliable, like careful reasoning or reliable testimony. A positive conclusion merely has epistemic ramifications for subjects who formed a belief 1–5 by relying on a faulty BFP and continue to depend on that faulty BFP. If subjects form a belief by means of a faulty BFP yet subsequently find other means of supporting that belief, their belief is not harmed.[9]

## 3. Belief-Forming Processes That Produce DCT-Beliefs

We now turn to belief-forming processes that produce beliefs 1–5. I discuss four recent explanations for why people tend to hold beliefs concerning a god-morality connection. Before discussing the explanations at length, I will start with some caveats.

A first caveat concerns the status of the theories discussed below. All are fairly new (last two decades) and all have been subject to criticism. Some have also questioned the scope and status of cognitive explanations in general.[10] Nonetheless, the theories are widely discussed among cognitive scientists and evolutionary theorists and received some empirical back-up. More testing and theorizing are likely needed to properly assess the reach and impact of all four theories. Nevertheless, for the remainder of this paper, I will proceed as if the theories are true or roughly true.

A second caveat is that the theories may not point to BFP's that *produce* beliefs 1–5 but rather processes that make such beliefs more attractive. Because of their evolutionary use or intuitiveness, some beliefs would be acquired more easily or exert a greater appeal to accept. Such features do not show how the beliefs were produced. Compare this to processes like peer pressure or prestige bias. Sometimes people find themselves more drawn to beliefs because a lot of members from their peer group or people with high social standing affirm them. These factors do not explain how a given belief is produced in a subject. Other BFPs, like reasoning or testimony, are likely responsible for the production of the beliefs themselves. It rather shows why some beliefs are more attractive. Peer pressure and prestige bias nonetheless still explain why a belief is held in a significant way. Both are, therefore, still epistemically relevant. I will also largely ignore this caveat in what follows. The processes discussed in all four theories might merely point to cognitive attractors rather than BFP, but this likely makes little difference in their epistemic import.[11]

### 3.1. Broad Supernatural Punishment

The first theory I consider is known as 'broad supernatural punishment'.[12] Defenders argue that the belief in moralizing, punishing gods (or rather BFPs producing those) was selected by natural selection. This belief makes people less keen to free ride and more likely to cooperate.

Human societies rely more heavily on a division of tasks than most other animal communities.[13] Human societies need to delegate tasks like hunting, foraging, and educating children to sustain themselves and flourish.[14] The division of labor is both humanity's greatest strength and weakness. A weakness is that this cooperation opens up the possibility of free riders, i.e., individuals who reap the benefits of work by others and do not contribute anything themselves. A large number of free riders in a society puts more load on non-free riders. It can also diminish the trust that is needed to keep cooperation going.

Belief in moralizing and punishing gods can help reduce the number of free-riders and generally make humans more eager to cooperate. If gods are seen as caring deeply about moral norms and punishing or rewarding humans in accordance with their moral behavior, the odds increase that humans will follow the norms. If those norms include social norms, like 'one ought to cooperate', the belief in moralizing gods has evolutionary beneficial effects on cooperation.[15]

The first theory does not primarily point to a BFP that produces belief in moralizing gods but rather why such a BFP is evolutionarily adaptive. Often the BFP itself is left unspecified or defenders refer to other more proximate explanations for religious beliefs. For our purposes, it suffices that if the theory is true, there is a BFP that produces belief in moralizing gods, which has an adaptive function.

### 3.2. Big Gods Theory

A second theory is highly similar to the broad supernatural punishment theory. Like the previous theory, defenders argue that the belief in moralizing and punishing gods (called 'Big gods') makes humans more prone to cooperate and is, therefore, evolutionarily adaptive. Unlike defenders of the previous theory, however, they do not claim that the belief (or BFPs responsible for the belief) is a biological adaptation, but rather a cultural adaptation.

Defenders of the big gods theory argue that belief in gods (moralizing or not) was not evolutionarily adaptive for most of human history.[16] For most of human history, groups were sufficiently small to allow for monitoring by institutions like the family or chiefdoms. Belief in moralizing gods gave societies an edge because they allowed humans to live in large-scale societies in the late Neolithic age. At that time, groups with beliefs in moralizing gods proved more successful than groups with different religious beliefs and gradually outcompeted them.

Like the previous theory, the big gods theory does not primarily explain how belief in moralizing gods is produced. It merely shows that a propensity to form such beliefs grew to be advantageous at some point in human history. The theory does seem to imply that a BFP that produces belief in moralizing gods arose at some point in human history and was selected for its evolutionary use.

### 3.3. Moral Dyad

A third theory is known as 'the Moral Dyad'.[17] Defenders argue that human moral reasoning intuitively takes a dyadic structure. When humans encounter a morally significant event (e.g., a crime or help), they intuitively look for a moral patient (the receiver of good or evil) and a moral agent (the giver of good or evil). Being dyadic, mixed forms where a subject is both a moral agent and a patient at the same time are rare. Moral evil, whether given or received, is usually cashed out as harm. In crimes, a moral agent harms a moral patient.

In most cases, it is not difficult to pinpoint the moral agents and patients. In other cases where harm is clearly involved, a moral agent seems absent. Clear examples are natural disasters. There are clear moral patients, usually humans who lost their lives or homes. Who the agent was that brought the harm about is unclear. In these cases, humans would intuitively connect the morally significant event to an invisible, divine moral agent. Natural disasters are easily regarded as punishments for transgressions. Morally good events without clear moral agents, like exceptional harvests, are easily seen as divine rewards.

Unlike the previous two theories, the moral dyad does point to a clear BFP. The moral dyad produces beliefs about a moral agent, which sometimes is God. Defenders do not discuss the evolutionary roots of the BRP at length.

### 3.4. Costly Signalling

A final theory relevant to our discussion is the costly signaling theory.[18] The theory bears some similarities to the first two theories but is primarily concerned with explaining ritual behavior.

Defenders of costly signaling theory argue that religious rituals are means to signal one's commitments to prosocial norms. By engaging in hard-to-fake performances, like religious rituals, people show their commitment to norms dominant in the group. Engaging in such practices reduces a subject's fitness in the short run because it requires considerable investment of time, resources, and energy. They increase fitness in the long run because they serve a similar purpose like belief in moralizing gods in theory 1 and 2. Signaling serves to weed out free riders and increases overall cooperation.

## 4. Debunking?

We can now turn back to the central question and investigate whether the theories discussed in the section above signal bad epistemic news for beliefs in line with DCT. Before we can do so, we need a clear idea of which beliefs all the theories explain.

### 4.1. Broad Supernatural Punishment

Which belief does the first theory, broad supernatural punishment, explain? If true, it explains,

- Why people believe that gods care about morality.
- Why people believe that gods punish or reward people in accordance with their moral behavior.

For both beliefs, the explanation is that doing so increases fitness. Strictly speaking, the theory does not explain how both beliefs are formed, but rather why the belief (or BFP that produces the belief) was retained by natural selection. A defender could add that the BFP that produces both beliefs was randomly thrown up by genetic mutations and proved more adaptive than other mutations.

Of both beliefs, only (a) is among the beliefs we distinguished in Section 2. Belief (a) fits well with weaker forms of DCT, like (3) and (4).

Does the theory show that beliefs like 3 and 4 are formed in an epistemically bad way? In a related discussion,[19] Paul Griffiths and John Wilkins argued that beliefs formed by BFPs that were selected for adaptive reasons are epistemically suspect[20]. Problems arise if the BFP is selected for reasons other than truth (Wilkins and Griffiths 2012). According to defenders of broad supernatural punishment, the BFP that produces belief in moralizing gods was not selected because such gods actually exist but because believing they do benefits cooperation. The BFP would, therefore, have been selected regardless of whether the beliefs it produces are true. The insensitivity towards truth is a reason not to see the BFP as a reliable guide to reality.

Commonly, defenders who reply to debunking arguments that rely on evolutionary explanations do so in terms of theistic evolution. The argument only works if the BFP was randomly thrown up by genetic mutations and selected for fitness reasons only. If

evolution is guided and planned by a divine being, the BFP could be a willed creation. A divine being can have various reasons to have a BFP that produces beliefs about moralizing gods that evolve in humans. He may want people to know that he cares about morality. He may also want to increase the odds that humans follow moral norms. To achieve this goal, he may want to make sure that believing in moralizing gods increases fitness and is, therefore, selected. On theistic accounts of evolution, the fact that a BFP seems insensitive to the truth with regard to its adaptive value, therefore, need not cause a problem.

Without theistic evolution, things seem bleak for belief in moralizing gods produced by a BFP selected to increase cooperation. The belief seems little more than a tool to aid fitness like other false beliefs like an augmented sense of one's own abilities.

Additional reasons in defense are made. Supporters of different religious traditions often present additional reasons to defend the idea that God is somehow connected to morality or that moral norms are aligned with God's command. In Christianity, Judaism, and Islam, adherents gain support for their beliefs from revelations. God is believed to have revealed himself as morally concerned. He does so by giving moral rules (e.g., the 10 commandments) and by punishing or rewarding people (e.g., Gen. 19:1–28). When adherents are prompted to answer why they believe that God is morally concerned, they are far more likely to refer to accounts of revelation than to a general sense that God is morally concerned, which is unknowingly produced by a BFP.

Adherents of other traditions like African indigenous religions or East Asian traditions often refer to visions reported by mediums or members of the group. In these visions, gods reveal taboos or moral norms. Members also report actual punishments or rewards for moral transgressions by gods.

To successfully debunk the belief, the debunker needs to show that referring to revelations or religious experiences is nothing more than confabulations on a vague sense or belief produced by an evolved BFP. This remains a difficult and arduous task.[21]

### 4.2. Big Gods Theory

Not much is different if the big gods theory turns out to be true. Belief in moralizing gods remains a product of natural selection (in this case cultural evolution). The belief is also selected for reasons different from the truth. Therefore, people would find themselves believing in moralizing gods whether the belief was true or not.

A defense in terms of theistic evolution is available as well. Most accounts of theistic evolution focus on biological evolution (e.g., Newman 2003). Defenders argue that God arranges the initial state and parameters of the evolutionary process. Some add that God can intervene to direct the evolutionary process at a later stage as well. There is no good reason to think that the same cannot hold for cultural evolution as well. Here, God can intervene by directing human behavior or activity.

One can object that an appeal to theistic evolution is ad hoc. Claiming that beliefs concerning a close link between God and morality are justified because they are brought about (indirectly) by God is sometimes considered an intellectual cop-out. All outputs of evolution (biological or cultural) can be regarded as willed by God and, therefore, justified or on track. Resorting to theistic evolution also provides no means of distinguishing reliable from unreliable evolved BFPs. Discussing the arguments for and against theistic evolution lies beyond the scope of this paper. A resort to theistic evolution in this discussion is, however, legitimate because the nature of evolution (guided or unguided) is not part and parcel of standard evolutionary theory. As many have noted, current evolutionary biology allows it to be framed in a guided and unguided view of the direction of evolution.[22] Insisting that evolved BFPs were the result of processes merely aimed at survival or reproduction also requires additional arguments.

Explaining belief in moralizing gods by pointing to the big gods theory also faces the challenge of explaining other reasons put forward in defense, like revelations and experiences.

### 4.3. Moral Dyad

Are matters different if the moral dyad theory turns out to be true? Considering the moral dyad as a BFP explains why people form the belief that a given morally significant event was caused by God. Enough of these may give rise to the belief that God is concerned with morality. Like the two theories above, the moral dyad does not explain why people form the belief that God's commandments lie at the root of morality.

Defenders of the moral dyad theory do not dwell on its evolutionary origins. Evolutionary insensitivity is, therefore, not a problem. The moral dyad may lead to epistemic worries for other reasons. Defender Kurt Gray suggests that the moral dyad is making a mistake when it concludes with a divine moral agent (Gray and Wegner 2010).[23] In 'normal' cases where a moral agent is clearly identifiable, the moral dyad is functioning well. It overreaches when it tries to identify a moral agent for large-scale events like natural disasters.

The suggestion that the moral dyad is making a misattribution seems to assume that there is no moral agent responsible for events like natural disasters. On naturalistic views, events like natural disasters are nothing more than the result of blind natural processes like changes in air pressure or tectonic shifts. Being non-agential, these do not qualify as moral agents. However, in most theistic traditions, an additional causal layer is sometimes added to these natural causes. God is sometimes regarded as the prime cause who sets events in motion by means of secondary causes like natural events.[24] Many contemporary theologians and believers may feel a certain unease in seeing God as the cause of morally bad events like natural disasters. Traditionally, however, the attribution of morally bad events to God was not uncommon. Several theodicies can also accommodate God using morally bad events for some greater purpose like soul-building.[25] A similar secondary cause account can be given for morally good events like unexpected healings or fortunes.

Pointing to the operations of the moral dyad as BFP for beliefs in line with DCT, therefore, does not show that these are epistemically tainted. Concluding with God as a moral agent is only problematic if a certain view regarding the cause of large-scale moral events is taken on board, being that these have only natural causes. These are not shared by adherents of most theistic traditions. They are, therefore, likely not shared by those who hold beliefs in line with DCT.

### 4.4. Costly Signaling

We come to the final of the four theories. Costly signaling, if true, explains why people believe that religious belief (or rather religious practice) is required for proper moral behavior (belief 5)). Because engaging in religious rituals is seen as signaling one's commitment to a community's norms, ritual practice can easily be regarded as a requirement for being morally virtuous. It is possible that humans are morally virtuous without engaging in costly rituals, but they are harder to spot. This all can easily give rise to the belief that people who do not engage in religious rituals are free riders or in other senses not morally virtuous.

A similar argument I discussed in the sections on broad supernatural punishment and the big gods theory can be stated using costly signaling. Here, the mechanism responsible for connecting morality to belief in God was selected by evolutionary forces. In naturalistic interpretations of evolutionary theory, evolutionary forces primarily favor traits that improve survival or reproductive success; therefore, the mechanisms responsible for costly signaling would have been selected regardless of whether its outputs are true or false.

A defender could again reply in terms of theistic evolution. God could want people to have a mechanism to weed out free riders from the morally virtuous. Contrary to other beliefs, the belief that religious belief (or ritual practice) is required for proper moral behavior is less supported by additional reasons. People may report anecdotal evidence in favor of the claim, but it is far less supported than other beliefs in line with DCT.

Costly signaling also implies different epistemic worries for belief (5). As with many evolved mechanisms, the mechanism that connects costly ritual behavior to moral behavior

is imprecise. Many evolved mechanisms appear to operate on a better safe than sorry strategy.[26] If the theory is true, rituals serve as a means to signal adherence to moral norms. Signaling is just signaling. Some humans might not have the need to signal moral virtue because they have no need for social capital or other benefits. This does not show that they are not morally virtuous. Solely relying on signaling to decide who is morally virtuous and who is not, therefore, will only reveal morally virtuous humans who also signal their virtue. Non-signaling virtuous people are left out.

Relying on signaling behavior to form beliefs about who is morally virtuous and who is not will thus lead to mistakes. The BFP at work here is not very precise or reliable. Beliefs formed in this way therefore are epistemically tainted.

## 5. Conclusions

The discussion in this paper revealed no significant worries for beliefs in line with the divine command theory if theistic evolution is accepted. The costly signaling theory raises worries about the belief that religious belief is required for proper moral behavior. However, this belief is not central to the divine command theory and is often disavowed by defenders.

Other beliefs connected to the divine command theory, especially those in line with weaker versions, can only be salvaged by appealing to theistic evolution. Defending them requires a strong view of theistic evolution wherein God intervenes in the evolutionary process by having particular belief-forming mechanisms evolved. Averting the threat from the seemingly unreliable operations of the moral dyad requires a view where God causes natural disasters or other large-scale morally significant events. This likely goes beyond thinner versions where God sets the initial parameters for evolution to take place in.[27]

**Funding:** This research received no external funding.

**Institutional Review Board Statement:** Not applicable.

**Informed Consent Statement:** Not applicable.

**Data Availability Statement:** Not applicable.

**Conflicts of Interest:** The author declares no conflict of interest.

## Notes

[1]　I exclusively focus on recent defenses of the divine command theory. The theory is of course much older. See for example: (Augustine 2005).

[2]　Well-known examples are the Christian and Jewish Ten Commandments and the Hindu Dharma. Religious conceptions of morality where moral rules stem from taboos issued by spirits or ancestors may also be regarded as examples of DCT.

[3]　Anscombe contrasts a law-based conception of morality (or deontological conception) with utilitarian and virtue-based conception of morality. Anscombe argues for the superiority of the latter (Anscombe 2020).

[4]　Tian is commonly regarded not as a material natural phenomenon but as a divine entity which is the proper object of worship by the Chinese.

[5]　Disavowal is common in defenses of the need of God's existence for objective morality. See for example: (Craig and Wielenberg 2020).

[6]　See (Tamir et al. 2020) for more numbers broken down by country.

[7]　For an overview, see: (Austin 2006).

[8]　Defenders of most internalist epistemologies (cf. Madison 2010) would require that the subject knows that the BFP is distorting reality in order to lose justification. Discussion on the requirement for such meta-beliefs for justification are beyond the scope of this paper.

[9]　For a defense of justification by resorting to other reasons or other means of support, see: (McBrayer 2018).

[10]　See, for example: (Oviedo 2018).

[11]　The immediate causes for belief-formation, like reasoning or testimony, may rectify the epistemic shortcomings of attractors. If attractors are shown to be misleading, yet reasoning or testimony is solid, beliefs are not tainted.

[12]　For defenses of the theory, see: (D. Johnson 2016; Bering and Johnson 2005).

[13]　Some animal communities, like bees or ants rely even more heavily on cooperation for their survival and procreation.

[14]    It occurs that a human individual does all these tasks him- or herself. Such occasions are however, rare and solo individuals usually have a much shorter lifespan.

[15]    Some defenders add that gods need to have access to intentions and thoughts to have a beneficial effect on cooperation (Bering and Johnson 2005).

[16]    Some argue that belief in gods arose as a by-product, a trait without intrinsic adaptive value which evolved along with other adaptive traits (Davis 2017).

[17]    The theory is primarily defended by Kurt Gray (Gray and Wegner 2010).

[18]    For defenses, see: (Soler 2012; Sosis and Bressler 2003).

[19]    See: (Van Eyghen and Bennett 2022) for an extended discussion of the argument.

[20]    Wilkins and Griffiths conclude that such beliefs ought to be believed with 'less confidence'.

[21]    The debate concerning justification by means of religious and revelatory experiences is vast with numerous arguments on both sides (see for example (Alston 1993; D. K. Johnson 2022). All this shows that additional arguments are needed to discard justification from experiences of this sort.

[22]    For examples, see: (Van Woudenberg and Rothuizen-van der Steen 2015; Plantinga 2011).

[23]    To avoid the mereological fallacy (Bennett and Hacker 2022), the terms 'making a mistake' and 'concludes' should be read functionally. The term 'making a mistake' means something like 'producing false information' and the term 'conclude' roughly 'deriving new information'.

[24]    For a discussion, see: (Hallanger 2016). For a recent criticism, see: (Kittle 2022).

[25]    For a defense of the soul-building theodicy, see: (Hick 2010).

[26]    A well-known example is the case of agency detection. Various authors argued that humans are too quick to detect agency on vague or limited evidence because doing so was evolutionary beneficial. See: (Atkinson 2023) for an overview of various accounts.

[27]    I thank an anonymous reviewer for suggesting this point.

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
