# Peer review of "Debunking Divine Command Theory"

_religions, doi:10.3390/rel14101252_

Round 1

Reviewer 1 Report

The context and limits of the study should be more fully explained, and the thesis should be more clearly stated at the beginning: the author there simply proposes to assess the theories. The conclusion says that the criticisms considered present "no significant worries for beliefs in line with the divine command theory." This could be both stronger and more specific. It could be a valuable contribution to answer certain criticisms of divine command theories, but the author has more work to do in making this a compelling answer. 

Editing for style and grammar is needed. There are numerous errors of spelling and grammar. Some of the transitions are awkward. 

Author Response

Dear sir/ dear miss,

I added some more details in the introduction and abstract to make clear what thesis I'm defending. I also discuss the limitations in section 3.

Best regards,

Author

Reviewer 2 Report

Line 54 (and 65) – where is the evidence that most defenders of DCT reject the notion that belief in God is necessary to behave morally?

Line 58 – There needs to be a comma in the number.

Line 69 – so from here on out will you be rejecting the idea that this notion should be included in DTC? While I found this section very interesting, I’m not sure of the relevance (at this point) to the overall thesis. Clarify why this information is presented.

Line 88 – capitalize “on”

Line 92 – I don’t think the plural of ratio is ratio’s. it’s just ratios.

Line 93 – it should be “is”…IS looking with beer googles. …. But is “beer googles” really what they call what they are talking about? “Beer googles” is also a term for how “unattractive women” look to drunk men. “She wasn’t my type, but I was drunk, and she looked good with ‘beer googles’ on.” You need to clarify that you are talking about actually looking out of glasses that are made of the bottom of beer bottles—unless they actually do mean “being drunk,’ in which case you need to clarify that too.

Line 104 – I am inclined to think this matters for those defending DTC philosophically too, because the phenomenon of constructing philosophical arguments for pre-drawn conclusions is quite common—even among professional philosophers. Yes, they have a philosophical argument—but the only reason they constructed such an argument (and likely the only reason they think it is good) is because they already accepted the conclusion. If such a person realizes this, and realizes that the initial (non-philosophic) reasons they accepted DCT were faulty, such a person might realize that their reasoning in its favor its “motivated,” and that that motivation has blinded them to the insufficiency of their argument, and blinded them to the success of the objections.

line 112 – capitalize “more”

Line 113 – a “Nevertheless” would be good to start the last sentence of the paragraph. “Nevertheless, for the remainder of this paper…”

Line 129 – edit: The first theory I wish to consider is known as …

134 – add “and” before “educating” (and after the oxford comma).

155 – a “however” before “they” would aid in readability.

161 – “Then” should be “At that time.”

163 – “those” should be “them.” “…outcompeted them.”

189 – “a final theory” not theories.

196 – oxford comma is missing

201 – “in the section above” –the “the” is missing. … also.. “…all those theories explain” …the “those” is missing.

216 – Capitalize “in” after the question mark.

239 – oxford comma is missing after Judaism

252 – No, to show that the “religious experience” objection doesn’t work, one need only show that beliefs produced by religious experience or visions by mediums are not justified—that such processes do not reliable produce true belief (that they are not reliable BFPs). This has already been done fairly successfully. See Johnson, David Kyle. “Why Religious Experience Can’t Justify Religious Belief.” SHERM (Socio-Historical Examination of Religion and Ministry) (Vol. 2, No. 2, Fall 2020): 26-46. https://doi.org/10.33929/sherm.2020.vol2.no2.03

260 – The ad hoc nature of the theistic evolution response needs to be addressed. It’s just basically saying “Well, what if God did it; it would be justified then.” If such an answer is intellectually acceptable, then basically no objection to any argument ever works, because you can just invoke God to get out of any objection. The Shourd of Turin doesn’t date to the 1st century, but instead the 13th. “Well what if God created in the 13th.” Or “What is God wanted to make it look younger to test our faith.” I feel like the intellectual “desperateness” of theistic evolution as a response to this problem needs to be addressed somehow.

269 – I think “explain” should be “explains.”

280 : The author says: “The suggestion that the moral dyad is making a misattribution assumes that there is no moral agent responsible for events like natural disasters.” No. “The conclusion that the moral dyad is making a misattribution is grounded in the argument that no moral agent is responsible for events like natural disasters”, and that argument is quite strong. First, science has proven beyond any doubt that the cause of such events (earthquakes, hurricanes, etc.) are natural conditions, like air pressure and tectonic shifts; this is not an assumption. That is why they are now called by philosophers “natural disasters” (and are only really called “acts of God” by insurance companies, and even then only as a convenient shorthand for “natural disasters.”) Again, this is not an assumption; it is as established as any scientific fact can be. What’s more, claiming that God is somehow responsible for the conditions that create such disasters (or the laws of physics that make them inevitable) is extremely problematic because (a) that makes God the author of evil, which by definition he cannot be and (b) adding God as an extra entity in this process is completely unnecessary and thus irrational (a violation of Ockham razor). And explanation just rooted in the laws is simpler than the “it’s the laws, but God causes the laws” explanation.

To boot, when natural disasters happen, and it comes to questions of why certain people were harmed and others were not, the best explanation is, without question, uncaused randomness. To think otherwise is intellectually bankrupt. As an example, suppose someone wins a bingo game; and then when they do, they stand up and declare “God chose me to win, and he chose all of you to lose.” While this might seem appealing to some “lay people,” intellectually honest philosophers could never defend such a thesis. When something general is bound to happen by probability—someone will win the bingo game, someone will be hurt in the hurricane—but the specific outcome (who will win or be harmed) is not determined (or indeterminate), the best explanation is that the specific outcome that does happen was due to chance. Invoking divine explanation may be tempting for various reasons to some, but it intellectually vacuous.

This fact needs to be pointed out in the paper.

284 – “added to these natural causes”…the “to” is missing.

291 – edit: “Concluding that God is a moral agent is only problematic if a certain view regarding the cause of large-scale moral events is accepted; but such views are not shared by most adherent of theistic traditions.”

Also, regarding the next sentence—the fact that those views are not shared by those who embrace DTC does not mean such BFPs aren’t epistemically problematic. They still are. Yes, you can add a “God did it” explanation onto of the already existing natural explanation, if you want to, and the problem goes away—but that move itself is not rational or reliable (usually such “God did it” explanations are shown to be unnecessary and untrue), so basically you are just replacing one epistemically problematic practice with another. … In other words, theists might say they are unbothered by this objection, because they just accept the “God did it” explanation for the BFP; but that is not a real solution if that move itself is just as problematic. It would be like showing someone direct evidence that the world is round, and them replying “I’m not bothered by that (and wont’ change my mind) because I think that evidence was planted by the lizard aliens that want you to think the world isn’t flat.” Yeah, if you really think that, you would be unbothered by the evidence I presented—but the move you made to get out of the problem is even worse.

312 – “Contrary than for” should be “Contrary to” I believe.

Conclusion: I think the worry does apply to DTC as a whole, not just belief 5. The only way out is invoking a particular variety of theistic evolution as a kind of ad hoc excuse to get out of the objection. (Notice that just believing God created the universe (and wrote its laws) doesn’t solve the problem—and that is a variety of theistic evolution; the version of theistic evolution that gets out of the objection has to have God stepping in at particular times to make certain BFPs work to fixation; that is much more akin to Behe’s intelligent design, which is pure pseudoscience. So I think the author could make the paper much more important and significant by broadening the thesis and saying that this problem really does pose a problem for DCT as a whole (not just belief 5). If the paper does this, I think it would be worthy of publication.

If, however, the whole goal of the paper was to say that these kinds of objections don’t really threaten divine command theory—if, in other words, the author is wanting to endorse the kind of theistic evolution “God did it” kind of ad hoc excuses to get out of the objection talked about here—as a way to say that DCT is not really threatened by this objection (and thus is unwilling to make such revisions)… Well, I don’t think that argument works at all, for the reasons I have articulated, and I would not advise publishing.

The more I think about it, I’m afraid that is what is going on here: the author is saying “these objections to DCT shatter then “you have to believe in God to be good” bit of DCT, but they don’t touch the rest of it. This is wrong. These objections to derail all 5 points of DCT; the paper could nicely point out why by showing that they only way to get out of the objection is to appeal to theistic evolution, which is problematic for other reasons and only an ad hoc excuse to get out of the objection. If the paper does that—yeah, that is an important contribution and a good argument. If, however, the goal is to say “theistic evolution can help the thesis get out of these objections”—no it can’t…at least, not in any kind of intellectually satisfying way. To think such an argument works would likely be the result of the kind of cognitive bias I spoke of before; the person has already concluded that DTC is true, and so they find this kind of ad hoc move satisfactory, when it is not.

One last thing to mention. The “argument from reason” –and its assumption that particular BFPs can be selected by natural selection—came to mind as I was reading this paper; it relies on erroneous assumptions about how evolution works that might be present in this paper. I can’t say anything more specific than that, but I encourage the author to look at debates on this argument to see whether further revisions might be needed (or at least to be ready for objections or discussion that might be leveled at this paper if it is published). There is a good discussion (a debate by two authors) on the Argument from Reason in Bassham’s “C.S. Lewis, Christian Apologetics” book; that debate the was continued in Philosophia Christi (Vol. 20, No. 2, 2019).

I put needed corrections in my previous comments. 

Author Response

Dear sir/ Dear miss.,

Thank you for your insightful comments.

I made minor textual changes as you suggested.

I added a note to exmplify that some defenders of DCT reject the need for belief to be moral.

I clarified why I proceed why the belief that reliosity is needed for morality is not included in DCT.

The point on theistic evolution addresses a major discussion in philosophy of religion that falls beyond the scope of this paper. I added a pragraph discussing the problem you raised in some more detail and why theistic evolution can be regarded as a good counterargument. Addressing the ad hocness of theistic evolution falls beyond the scope of this paper.

I emphasized the large role of theistic evolution in responding to the threat in the conclusion. I also added the need for a strong view of theistic evolution.

The issue of justification by means of religious experiences also falls beyond the scope of this paper. I added a footnote pointing to the ongoing discussion of this topic.

I changed the example of beer goggles to wearing bad glasses. This example better fits the line of argumentation.

I added some lines on arguments for previously held positions in philosophy.

I added that the claim that the moral dyad is making misattributions only works if natural events are seen as the result of blind forces. I added that this blindness or unguidedness is not part and parcel of standard scientific theories and takes on additional metaphysical positions.

I'm not sure how the argument from reason features into the arguments in the paper. The assumption that BFP's can be selected by natural selection is commonly accepted in evolutionary psychology and does not seem to rely on a naturalistic view or a view that nothing mental exists.

Reviewer 3 Report

While the paper is, overall, well written, I believe it needs a clearly stated thesis that controls the overall focus and provides purpose to the paper.  The research is interesting, and besides some very minor grammatical or syntactical errors, and minor issues with word choice. 

Lack of a clear thesis is, in my opinion, fatal.  The paper should also clearly state the thesis in the abstract, and in the initial portion of the paper.  The reader simply doesn't know what to expect from the paper, not does she know what might be learned from the paper.  This should happen very early.

I suggest rewriting the abstract to reflect a purpose and goal of the paper besides "assessment" of others' ideas.  While literature review is great, it should motivate a clearly stated thesis that controls the relevance of the content of the paper throughout.

While the introduction clearly states the the author intends to "discuss, discuss, and assess" this is not usually enough for a philosophy paper.  There needs to be a contribution to the literature, not simply a review of it, as thorough and enlightening as this is.  There should be an explicit argument.

Very good.  There are only very minor errors.  An obvious example is lines 201-202, where "all" should probably be "each."  Surely the author means "we need a clear idea of what each of the foregoing theories explain" (or perhaps entail, etc..).  And line 48, "A lot" should probably be "Many."  And in line 78, "from" should probably be "for."  There are other unfelicitous constructions occasionally throughout, which could be improved.  Overall, however, another close reading by someone other than the author should suffice to improve the readability well enough for publication.

Author Response

Dear sir/ dear miss.,

I added a clearer statement of the thesis i'm defending in the introduction and abstract.

I also phrased the introduction is less a 'discussion' but more argumentative.

Round 2

Reviewer 1 Report

An interesting study, with reference to the contemporary opinions on the connection between God and morality.

Overall, I think the author successfully shows some errors in common arguments against DCT. I think it could be a richer study if it were not limited to looking at arguments from the past two decades, that is, if it made reference to the incredibly rich tradition of Christian thinking. One cannot do everything, but I think even a passing reference to, say, Augustine's On the Advantage of Believing, would show the author's awareness that these are by no means new issues, even if they are cast in new terminology.

I suggest a few minor edits: 

p.2, ln. 46: 'civil laws' instead of 'legal laws'

p.2, ln. 61: 'origin of'

p.2, ln. 76: it would help to restate here the two positions for clarity

p. 3, ln. 89: 'for' in place of 'from'

p. 7, ln. 279: 'question-begging' in place of 'ad hoc' (?)

p. 8, ln. 312: 'prime' in place of 'prima'

p. 8, ln. 315: insert comma after 'traditionally' (or say 'Traditionally, however, ...)

p. 9, ln. 357: 'not very precise or reliable'

Author Response

Thank you for the comments.

I made most suggested changes apart from the one on 'ad hoc'. I believe the term ad hoc is better than question begging here because the idea is that theists just resort to theistic evolution to avoid any problem on their road.

I added a reference to St. Augustine.

Reviewer 2 Report

I believe your revisions are adequate. 

Author Response

No changes

Reviewer 3 Report

Improved.  Nice work.

Author Response

no changes